# Does probe-tube verification of real-ear hearing aid amplification characteristics improve outcomes in adult hearing aid users? A protocol for a systematic review

Ibrahim Almufarrij  ,[1,2] Kevin J Munro,[1,3] Harvey Dillon[1,4]

¹Manchester Centre for Audiology and Deafness, School of Health Sciences, The University of Manchester, Manchester, UK
²Department of Rehabilitation Sciencess, College of Applied Medical Sciences, King Saud University, Riyadh, Kingdom of Saudi Arabia
³Manchester University Hospitals NHS Foundation Trust, Manchester Academic Health Science Centre, Manchester, UK
⁴Department of Linguistics, Macquarie University, Sydney, New South Wales, Australia

**Correspondence to**
Ibrahim Almufarrij;
ialmufarrij@ksu.edu.sa

## ABSTRACT

**Introduction**  Using a probe-tube microphone to measure and adjust the real-ear performance of the hearing aid to match the prescription target is recommended and widely used in clinical practice. Hearing aid fitting software can approximately match the amplification characteristics of the hearing aid to the prescription without real-ear measurements (REMs), but using REM improves the match to the prescribed target. What is unclear is if the improved match results in a better patient-reported outcome. The primary objective of this review is to determine whether the use of REM improves patient-reported outcomes in adult hearing aid users.

**Methods and analysis**  The review's methods are in accordance with Preferred Reporting Items for Systematic Reviews and Meta-Analyses Protocols guidelines. MEDLINE, EMBASE, PsycINFO, CINAHL, Web of Science and CENTRAL via Cochrane Library will be searched to identify relevant studies. The review's population of interest will include adults with any degree of sensorineural or mixed hearing loss who have been prescribed with acoustic hearing aids. The included studies should compare REM fitting to the initial fit provided by the manufacturer's fitting software. Hearing-specific health-related quality of life is the primary outcome but secondary outcomes include self-reported listening ability, speech recognition scores, generic health-related quality of life, hours of use, number of required follow-up sessions and adverse events. Randomised and non-randomised controlled trials will be included. The risk of bias in the included studies will be evaluated using Down and Black's checklist. The quality of the overall evidence will be assessed using the Grading of Recommendations, Assessment, Development and Evaluations tool.

**Ethics and dissemination**  Ethical approval will not be sought because this systematic review will only retrieve and analyse data from published studies. Review results will be published in a peer-reviewed journal and presented at relevant scientific conferences.

**PROSPERO registration number**  CRD42020166074.

## Strengths and limitations of this study

► This review will be first to synthesise data that compare two commonly used approaches to fit hearing aids to prescription targets (ie, initial-fit and real-ear measurements).
► A variety of outcome measures will be used to compare these two approaches.
► The implications of this review are important for the emerging category of over-the-counter and direct-to-consumer hearing aids that do not involve real-ear measures.
► We will include only studies that compare these fitting approaches among adults with sensorineural or mixed hearing loss, because children and those with other types of hearing loss may require different amplification characteristics.
► Grey literature will not be included (ie, conference abstracts and theses).

## INTRODUCTION

Hearing loss is the most prevalent sensory deficit, affecting more than 5% of the world's population.[1 2] Untreated hearing loss reduces peoples' ability to communicate effectively, which could lead to social isolation, depression and decreased health-related quality of life.[3] Hearing loss is associated with an increased risk of cognitive decline and dementia but causality is unknown.[4] The rate of unemployment among individuals with hearing loss is higher than that of the general population, costing the UK economy about £25 billion annually in terms of lost economic output.[5]

The most prevalent type of hearing loss in adults is sensorineural loss, accounting for more than 90% of all adults with hearing loss in the UK.[6] The primary intervention for permanent hearing loss is the use of acoustic hearing aids.[7] These devices are designed to restore audibility of low-level sounds, maximise intelligibility of conversational level sounds and for loud sounds.[8] Hearing aids reduce the handicap caused by mild and moderate hearing loss (eg, decreased health-related quality of life).[9]

In the UK, hearing aids are fitted and prescribed by state-registered hearing health

professionals. The selection of the most appropriate hearing aid is based on a combination of audiometric and non-audiometric factors (eg, hearing threshold levels, type of hearing loss, physical capabilities of the patient and patient preference, respectively). The optimal amplification characteristics for each hearing aid user are specified according to validated prescription formulae (eg, National Acoustic Laboratories Non-Linear 2),[10] which takes into account the patient-specific data (eg, gender, language and type and degree of hearing loss) to personalise hearing aid output. Modern hearing aid software can approximate the amplification characteristics and the use of this approach for hearing aid fitting is known as initial fit. A real-ear measurement (REM), which is a tool that can be used to accurately measure the output of a hearing aid when it is coupled to the individual's ear via a soft probe-tube microphone,[8] can also be used to guide the adjustment of the output of the hearing aid, so that it matches the proposed amplification target that was provided by the validated prescription formula.

Numerous studies have shown that the amplification characteristics approximated by the hearing aid software are inaccurate and significantly deviate from prescription targets.[11–13] These studies also suggest that the use of REMs can improve matches to prescription targets. Given the available evidence, using REMs to verify the output of hearing aids and achieve better matches with prescription targets, has been endorsed by the British Society of Audiology[14] and most international hearing organisations (eg, American Speech-Language-Hearing Association).[15] Nevertheless, it remains unclear whether improved matching to prescription targets results in better outcomes.

Using REMs to match hearing aid output to prescription targets requires valuable clinical time, which could otherwise be used to deliver alternative health and hearing services (eg, identifying and addressing other hearing problems).

### Rationale
There is no published systematic review examining the evidence on whether the use of REM improves patient-reported outcomes in adult hearing aid users. Determining the effectiveness of hearing aids fitted using REM would help decision-makers and stakeholders either to continue recommending or to abandon routine use of REM during adult hearing aid fittings. This is an urgent knowledge gap for both stakeholders and clinicians, given that the cost of the REM system is relatively high and it consumes a large amount of clinicians' valuable time. The findings have implications for the emerging category of over-the-counter or direct-to-consumer hearing aids. If fitting hearing aids using REM does not benefit patients, then one of the potential obstacles to people fitting their own hearing aids is overcome. However, if it is found to be beneficial, decision-makers should continue recommending the use of REMs and poorer outcomes will be expected for direct-to-consumer hearing aids.

### Objectives
The objective of this review is to systematically evaluate the current evidence on whether the use of REMs to match the hearing aid's output to one of the validated prescription targets improves outcomes in adult hearing aid users.

## METHOD AND ANALYSIS
The protocol of this systematic review has been preregistered with the International Prospective Register of Systematic Reviews. The systematic review's methods are reported in accordance with Preferred Reporting Items for Systematic Reviews and Meta-Analyses Protocols (PRISMA-P) guidelines.[16]

### Eligibility criteria
The inclusion and exclusion criteria for studies are reported in accordance with participants, interventions, comparators, outcomes and study designs elements.

#### Participants
Participants will include adults (≥18 years old) with any degree of sensorineural or mixed hearing loss. Studies that report only a qualitative description of age and threshold of hearing will also be included. If both children and adults are included in a clinical trial, the study will be excluded unless the data were interpreted and reported independently. Participants with other types of hearing loss (ie, conductive or fluctuating hearing loss) will be excluded.

#### Interventions
The intervention that will be included comprises conventional acoustic hearing aids that are programmed to a prescription target using a REM system. Assistive listening devices, hearables, personal sound amplification products and direct-to-consumer hearing devices will be excluded. Implantable devices (eg, cochlear implants), bone conduction hearing aids or contralateral routing of sound hearing aids will also be excluded.

#### Comparators
The comparisons of interest are hearing aids that are programmed to the manufacturers' approximation of the wearers' hearing loss without REMs (ie, initial fit approach).

#### Outcomes
The primary outcome of interest is hearing-specific health-related quality of life (eg, Hearing Handicap Inventory for the Elderly).[17] Secondary outcomes of interest are self-reported listening ability (eg, Abbreviated Profile of Hearing Aid Benefit),[18] general benefit of hearing aids (eg, International Outcomes Inventory for Hearing Aids),[19] speech recognition in quiet or noisy environments, generic health-related quality of life, hours of use per day, number of required follow-up care sessions

(ie, for further fine-tuning) and adverse events (eg, noise-induced hearing loss).

## Study designs

Randomised controlled trials and non-randomised controlled trials will be included. Case reports, conference abstracts, book chapters, dissertations, theses, reviews and clinical guidelines will be excluded.

## Information sources

Studies that meet the aforementioned eligibility criteria will be identified using a systematic search strategy of the following databases: MEDLINE, EMBASE, PsycINFO, CINAHL, Web of Science and CENTRAL via Cochrane Library. No search restrictions will be applied in terms of the publication's language, status and year.

The reference lists of the included publications will be manually scanned to identify further studies. Using Google Scholar 'cited by' feature, publications that have cited any of the included studies will be screened to identify additional relevant articles. Prior to analysis, the searches will be repeated to identify any other relevant studies if there is a significant delay between searches and a manuscript's submission for publication (>6 months).

## Search strategy

The search protocol and methods will be developed by a medical information specialist from Systematic Review Solutions. The search terms will be based on experts' opinion, free text and controlled terms from Medical Subject Headings, Excerpta Medica Tree and CINAHL headings. The search terms will include many free text terms because truncations may not work well with phrases (eg, Hearing difficult* might not capture Hearing difficulties). The search strategies for all databases are reported in online supplementary appendix 1.

## Study records

### Data management

Search results, including titles, authors' details, publication years, publication journals and abstracts, will be extracted to EndNote V.X9 Reference Management software (Clarivate Analytics, 2018). The same software will also be used to remove any duplicates prior to the initial screening. Next, IA will export the title and abstract of all identified articles into an Excel spreadsheet, so that they can be easily screened against the eligibility criteria. The reason for any article's exclusion can also be documented and assessed by KM and HD. Each article will be assigned a unique number that is linked to the full details of the article.

### Selection process

The title and abstract of all identified studies will be screened independently by IA and KM to determine eligibility for inclusion. A further inspection will be used when there is a discrepancy between the two investigators; this will include assessing the full article. Any disagreement will be resolved by discussion and/or by consulting with

| Table 1 | Data items |
|---|---|
| **General information** | **Authors (year) title** |
| Method | Study design<br>Total study duration<br>Sequence generation<br>Sequence concealment<br>Blinding |
| Participants | Total number<br>Country/setting<br>Age<br>Sex<br>Inclusion and exclusion criteria |
| Intervention | Intervention group<br>Comparator group |
| Outcome | Primary outcome<br>Secondary outcome(s) |
| Funding source | |
| Declaration of interest | |
| Notes | |

the third author (HD). The full text will be retained and inspected by IA and KM for all articles that match the inclusion criteria. Any disagreements will be resolved via discussion or by consulting the third author. Following PRISMA recommendations,[20] a flow diagram will be used to present the study selection process.

## Data collection process and data items

IA and KM will independently extract the data from the included studies. Should any discrepancies arise, these will be resolved through discussion or consultation with the third author (HD). The data will be extracted in a predesigned data extraction form adapted from the Cochrane Handbook.[21] The extracted data will include but will not be limited to authors (year), methods, participants, intervention and outcomes (see table 1). Data presented on graphical forms will be extracted using an online extraction tool (eg, WebPlotDigitizer; https://automeris.io/WebPlotDigitizer) when necessary.

## Risk of bias in individual studies

The assessment of the risk of bias will be conducted independently by IA, KM and HD. Any disagreements will be resolved using a majority decision. Given the limited number of randomised controlled trials in the field of audiology, it is anticipated that most of the extracted studies will be non-randomised controlled trials; therefore, the Downs and Black[22] checklist will be used because it is easy to administer, has a well-established validity and reliability and can be used to assess the methodological quality of both randomised and non-randomised studies. Because knowledge of the clinically important differences in hearing aid outcomes is lacking, scoring for the final item (number 27) will be modified based on whether or not the power calculation was performed. That is, one

point will be awarded if the calculation was conducted and zero points if it was not. Consequently, the maximum score will be 28. Articles scoring 26–28, 20–25, 15–19 and <14 will be regarded as having excellent, good, fair and poor quality, respectively. The assessment of the risk of bias will be conducted by the primary and secondary authors independently.

### Data synthesis

A meta-analysis will be conducted if enough data are available (ie, more than one study). If the studies used the same continuous outcome measures, the mean differences will be calculated with their 95% CI. When different outcome measures are used, the standardised mean difference will be measured along with 95% CI. If the studies used similar dichotomous outcome measures, the risk ratio will be calculated. If the statistical heterogeneity is low, fixed-effect meta-analyses will be computed; otherwise, only the random-effect meta-analyses will be calculated. For each meta-analysis, the effect size estimate will be calculated using the generic inverse of variance. The effect estimate will be reported along with its 95% CI. Forest plots will be used to present these results. Asymmetrical distribution of continuous outcomes (ie, skewed data) will be assessed by subtracting the lowest possible value from the mean and then dividing it by the SD. A ratio below 2 or 1 either suggests or indicates a skewed distribution, respectively.[23] Skewed data will be transformed where possible to reduce the skew distribution. Randomised and non-randomised controlled trials will be synthesised separately.

If meta-analysis is not appropriate, the risk of bias of each individual study will be assessed and the findings will be systematically reported

### Assessment of reporting bias

The existence of publication bias will be visually inspected using a funnel plot of the study size or precision as a function of intervention effect estimates.

### Assessment of heterogeneity

The percentage of variability between studies' outcomes that is due to heterogeneity rather than random error will be computed using an $I^2$ statistics. Given that the absolute threshold of $I^2$ is meaningless, the results will be interpreted as either low (0%–40%), medium (41%–60%) or high (61%–100%) heterogeneity.

### Dealing with missing data

The corresponding authors will be contacted, where necessary, if any of the data are missing. Data that are only presented in graphical forms will be extracted using the detailed method explained in the section Data collection process and data items. In case the SDs are missing and cannot be obtained from the corresponding authors, they will be estimated from the available data (eg, 95% CI and SEs). The reasons for the missing data will be assessed to identify whether they are missing at random or not.

### Subgroup analysis

When possible, subgroup analysis will be performed to determine whether age (ie, ≤55 or >55 years), severity of hearing loss and experience with hearing aids (ie, first-time vs experienced users) are possible sources of heterogeneity.

### Confidence in cumulative estimate

The quality of each outcome measure will be rated as high, moderate, low or very low using a well-developed assessment tool, the Grading of Recommendations, Assessment, Development and Evaluations tool (GRADE).[24] The quality rating assigned to each outcome measure reflects our level of confidence as to the veracity of the drawn estimate effect. The GRADE tool takes into account five principal domains: study limitations (eg, blinding and allocation concealment), inconsistency (eg, variabilities in effect size), indirectness (eg, differences in population), imprecision (eg, broad CI) and publication bias (eg, selective reporting of positive outcomes). Randomised controlled trials without serious shortcomings will be rated as high-quality evidence. That is, our confidence level is high enough to conclude that the true effect is close to the drawn estimate effect. Conversely, non-randomised control trials (eg, observational studies) will be assigned a low rating. In other words, the estimated effect may vary considerably from the true estimate of the effect. However, the assigned rating to randomised and non-randomised controlled trials could be subjected to either upgrading or downgrading by either one or two points on the basis of the seriousness of the above-mentioned assessment domains. A thorough discussion of these factors can be found in Schünemann *et al*.[25] IA and KM will carry out the assessment independently and disagreements will be resolved by discussion and/or by consulting HD.

### Sensitivity analysis

Sensitivity analysis will be administered to assess the robustness of the pooled estimates. That is, studies with a very high risk of bias (ie, those that scored less than 14 points on the Downs and Black checklist) will be removed from the quantitative synthesis.

### Patient and public involvement

Patients and the public were not formally involved in the development of the research question and outcomes. The findings will be displayed on the website of The Manchester Center for Audiology and Deafness. In addition, they will be shared with service user groups within the National Health Service and Action on Hearing Loss, the largest UK charity for deafness.

## ETHICS AND DISSEMINATION

Ethical approval will not be sought because this systematic review will only retrieve and analyse data from published studies. Review results will be published in a peer-reviewed journal, presented at relevant scientific conferences and disseminated on social media.

Dillon @HarveyDillon3

**Acknowledgements** IA, KM and HD are supported by the NIHR Manchester Biomedical Research Centre. The first author is also supported by the Deanship of Scientific Research at the College of Applied Medical Sciences Research Center at King Saud University.

**Contributors** IA is the guarantor of this review. IA developed and prepared the review protocol. The search procedure has been developed by a medical information specialist from Systematic Review Solutions. KM and HD contributed to the development of inclusion and exclusion criteria, search strategy and assessment of the risk of bias. KM and HD also critically reviewed and provided feedback on earlier versions of the protocol. This manuscript has been approved by all contributors.

**Funding** NIHR Manchester Biomedical Research Centre (funder reference: IS-BRC-1215-20007).

**Competing interests** None declared.

**Patient consent for publication** Not required.

**Provenance and peer review** Not commissioned; externally peer reviewed.

**ORCID iD**
Ibrahim Almufarrij http://orcid.org/0000-0003-4043-7234

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
