## [Reviewer comments · BMJ Open]

ARTICLE DETAILS

TITLE (PROVISIONAL)	Does probe-tube verification of real-ear hearing aid amplification characteristics improve outcomes in adult hearing aid users? A protocol for a systematic review
AUTHORS	Almufarrij, Ibrahim; Munro, Kevin; Dillon, Harvey

VERSION 1 - REVIEW

REVIEWER	Derek Hoare University of Nottingham, UK
REVIEW RETURNED	22-Mar-2020

GENERAL COMMENTS	Thank you for the opportunity to review this protocol for a systematic review comparing hearing aid fitting with and without the use of real ear measurement. The protocol is clear and consistent with the associated PROSPORO registration. I have minor comments for the authors to consider: P4L31 – here and elsewhere suggest using the term Hearing-specific health-related quality of life consistently. P5L24 – perhaps refer to grey literature rather than non-peer reviewed. These are peer reviewed so could be included. For completeness, and given the likely sparsity of data for this review, I would also consider including or otherwise reporting completed but unpublished trials that have been identified from trial registers. P5L58 – what ‘decision’ is referred to here? To use REM or not, recommend a hearing aid or not? If the latter then perhaps consider reference to shared decision making. P6L46 Rationale P6L55-58. This is useful information but the introduction feels that it is missing a more comprehensive description and evidence of these issues. For balance it would also be useful to consider the consequences of finding that use of REMs is superior for clinical practice and over the counter sales (currently reads a little that it is foregone that REMs will emerge as not needed). P7L24 – ‘are’ rather than ‘will be’ P8. Outcomes: please specify primary and secondary outcomes as per PROSPORO.
---

	P8L17 – composite self-report measures not an outcome, suggest ‘general benefit of hearing aids’. P8L52 – remove ‘e.g.’ Standard is to update your search is more than 6 months old at time of submission. P9. Data management section is confusing as first two line relate to screening; these are also duplicating later text so should be removed. Note also that PROSPORO refer to use of online data management tool such as Covidence. For consistency suggest being explicit what is benign used. P9L16 – here and later, would remove initials in brackets, not needed. P9. Process for full text screening needs to be described fully; screeners, dealing with discrepancies etc. P11L3-5 some grammar issues in these lines. P11. If risk of bias is assessed by two reviewers how will the data be managed? Take an average of score? Or discuss and agree single assessment rating for each study and each item? Involve third reviewer in disagreements also? P11L18 – do you mean ‘the same’ rather than ‘similar’? If ‘similar’ what criteria will be applied to deciding what is sufficiently similar to combine? How will synthesis be managed with respect to RCTs and NRCT? Separate meta-analyses presumably? Any adjustment to meta-analyses based on study quality? Subgroup analyses should be planned and conducted rather than depend on heterogeneity. Sensitivity analyses should be used to explore sources of heterogeneity, e.g. removing studies involving participants with characteristic x.
--	--

REVIEWER	Francesco Gazia Department of Adult and Development Age Human Pathology “Gaetano Barresi”, unit of Otorhinolaryngology, University of Messina Via Consolare Valeria 1, 98125, Messina ME, Italy.
REVIEW RETURNED	29-Apr-2020

GENERAL COMMENTS	The authors sought to create a protocol on the use of Real Ear and probes to verify a better functional outcomes in patients with hearing aids. The purpose of the work is very interesting, a review with meta-analysis could be of great contribution to the scientific community and facilitate the use of REM in each operating unit of otolaryngology or audiology. I would invite the authors to include only the manuscripts that describe how REM measurements were made, based on the angle (azimuth), in order to standardize the procedure. I would add a paragraph regarding the type of dome used in hearing
---

	aids. Based on the type of deafness, it is preferred to use various types of dome (Open, Tulip, closed, double closed ...); this section could be very important for the Forster Plot, because the REUG or REOG values may vary. Depending on the type of dome used, multiple statistical studies should be performed. I would invite the authors to also include SCOPUS in the database. For a correct use of the domes, I would like to refer and add this manuscript to references. Gazia F, Galletti B, Portelli D, Alberti G, Freni F, Bruno R, Galletti F. Real ear measurement (REM) and auditory performances with open, tulip and double closed dome in patients using hearing aids. Eur Arch Otorhinolaryngol. 2020 Feb 1.
--	--

VERSION 1 – AUTHOR RESPONSE

Reviewer: 1

Thank you for the opportunity to review this protocol for a systematic review comparing hearing aid fitting with and without the use of real ear measurement. The protocol is clear and consistent with the associated PROSPORO registration. I have minor comments for the authors to consider:

P4L31 – here and elsewhere suggest using the term Hearing-specific health-related quality of life consistently.

Thank you for your comment. The term ‘hearing-specific health-related quality of life’ has now been used throughout the manuscript (e.g., in the abstract on lines 16–18; on page 3, lines 16–26; and elsewhere).

P5L24 – perhaps refer to grey literature rather than non-peer reviewed. Theses are peer reviewed so could be included. For completeness, and given the likely sparsity of data for this review, I would also consider including or otherwise reporting completed but unpublished trials that have been identified from trial registers.

Thank you. The term ‘non-peer reviewed’ has been replaced with ‘grey literature’ in the modified manuscript (page 3, line 12).

P5L58 – what ‘decision’ is referred to here? To use REM or not, recommend a hearing aid or not? If the latter then perhaps consider reference to shared decision making.

Thank you. This sentence has been modified in the revised manuscript to read as follows (page 3, line 28):

‘The selection of the most appropriate hearing aid is based on a combination of audiometric and non-audiometric factors (e.g., hearing threshold levels, types of hearing loss, physical capabilities of the patient and patient preference, respectively).’

P6L46 Rationale

Thank you. The word ‘rational’ has been replaced with ‘rationale’ (page 4, line 22).

P6L55-58. This is useful information but the introduction feels that it is missing a more comprehensive description and evidence of these issues. For balance it would also be useful to consider the consequences of finding that use of REMs is superior for clinical practice and over the counter sales (currently reads a little that it is foregone that REMs will emerge as not needed).

Thank you. We are planning to discuss this point later on in the discussion section. For balance, we have added the following section (page 4, line 31 and page 5, lines 1–2):

‘However, if it is found to be beneficial, decision-makers should continue recommending the use of REMs and poorer outcomes will be expected for direct-to-consumer hearing aids.’

P7L24 – ‘are’ rather than ‘will be’

Thank you. We have replaced ‘will be’ with ‘are’ in the modified manuscript (page 5, line 10).

P8. Outcomes: please specify primary and secondary outcomes as per PROSPORO.

Thank you. The primary and secondary outcomes have now been specified in the manuscript (page 6, lines 5–8).

P8L17 – composite self-report measures not an outcome, suggest ‘general benefit of hearing aids’.

Thank you. We have replaced ‘composite self-report measures’ with ‘general benefit of hearing aids’ in the modified manuscript (page 5, lines 7–8).

P8L52 – remove ‘e.g.’ Standard is to update your search is more than 6 months old at time of submission.

Thank you. We have modified the sentence as suggested in the revised manuscript. It now reads as follows (page 6, line 25):

‘Prior to analysis, the searches will be repeated to identify any other relevant studies if there is a significant delay between searches and a manuscript’s submission for publication (> 6 months).’

P9. Data management section is confusing as first two line relate to screening; these are also duplicating later text so should be removed. Note also that PROSPORO refer to use of online data management tool such as Covidence. For consistency suggest being explicit what is benign used.

Thank you. We have modified the entire section. It now reads as follows (page 7, lines 11–17):

‘Search results, including titles, authors’ details, publication years, publication journals and abstracts, will be extracted to EndNote X9 Reference Management software (Clarivate Analytics, 2018). The same software will also be used to remove any duplicates prior to the initial screening. Next, IA will export the title and abstracts of all identified articles into an Excel spreadsheet so that they can be easily screened against the eligibility criteria. The reason for any article’s exclusion can also be documented and assessed by KJM and HD. Each article will be assigned a unique number that is linked to the full details of the article.’

P9L16 – here and later, would remove initials in brackets, not needed.

Thank you. We have removed all initials in brackets in the modified manuscript.

P9. Process for full text screening needs to be described fully; screeners, dealing with discrepancies etc.

Thank you. A description of the full process has been added to the modified manuscript as suggested. It now reads as follows (page 7, lines 23–25):

'The full text will be retained and inspected by IA and KJM for all articles that match the inclusion criteria. Any disagreements will be resolved via discussion or by consulting the third author.'

P11L3-5 some grammar issues in these lines.

Thank you. We have revised these lines to read as follows (page 9, lines 5–8):

'Because knowledge of the clinically important differences in hearing aid outcomes is lacking, scoring for the final item (number 27) will be modified based on whether or not the power calculation was performed. That is, one point will be awarded if the calculation was conducted and zero points if it was not.'

P11. If risk of bias is assessed by two reviewers how will the data be managed? Take an average of score? Or discuss and agree single assessment rating for each study and each item? Involve third reviewer in disagreements also?

Thank you for your question. This process was specified at the beginning of this section. However, we have further clarified the process as suggested. It now reads as follows (page 8, lines 10–11):

'The assessment of the risk of bias will be conducted independently by IA, KJM and HD. Any disagreements will be resolved using a majority decision.'

P11L18 – do you mean 'the same' rather than 'similar'? If 'similar' what criteria will be applied to deciding what is sufficiently similar to combine?

Thank you for your question. The word 'similar' has now been replaced with 'the same' (page 9, line 14).

How will synthesis be managed with respect to RCTs and NRCT? Separate meta-analyses presumably?

Thank you for your question. We intended to analyse RCT and NRCT separately. To that end, the following sentence has been added to the modified manuscript (page 9, lines 25–26):

'Randomised and non-randomised controlled trials will be synthesised separately.'

Any adjustment to meta-analyses based on study quality?

Thank you for your question. We will perform a sensitivity analysis if the quality of any of the included studies is found to be very low. To that end, the following lines have been added to the modified manuscript (page 11, lines 11–114):

'Sensitivity analysis will be administered to assess the robustness of the pooled estimates. That is, studies with a very high risk of bias (i.e., those that scored less than 14 points on the Downs and Black checklist) will be removed from quantitative synthesis.'

Subgroup analyses should be planned and conducted rather than depend on heterogeneity.

Thank you. We have modified the sentence to read as follows (page 10, lines 17–19):

'When possible, subgroup analysis will be performed to determine whether age (i.e. ≤ 55 or > 55 years), severity of hearing loss and experience with hearing aids (i.e. first-time vs. experienced users) are possible sources of heterogeneity.'

Sensitivity analyses should be used to explore sources of heterogeneity, e.g. removing studies involving participants with characteristic x.

Thank you. A sensitivity analysis section has been added to manuscript (page 11, lines 9–12).

Reviewer: 2

The authors sought to create a protocol on the use of Real Ear and probes to verify a better functional outcomes in patients with hearing aids. The purpose of the work is very interesting, a review with meta-analysis could be of great contribution to the scientific community and facilitate the use of REM in each operating unit of otolaryngology or audiology. I would invite the authors to include only the manuscripts that describe how REM measurements were made, based on the angle (azimuth), in order to standardize the procedure. I would add a paragraph regarding the type of dome used in hearing aids. Based on the type of deafness, it is preferred to use various types of dome (Open, Tulip, closed, double closed ...); this section could be very important for the Forster Plot, because the REUG or REOG values may vary. Depending on the type of dome used, multiple statistical studies should be performed.

Thank you for your comment. We appreciate this point, but it is highly unlikely that these variables will alter the effect size and confidence interval, as it is common for studies to control for these variables and use similar types of domes (or moulds) for both the intervention and control groups. That is, this variable will likely have little or no effect on the effect size and its confidence interval. If, by chance, we find that a study uses different domes (or moulds) for intervention and control groups, we will discuss it in the discussion section and include the reference you have listed below.

I would invite the authors to also include SCOPUS in the database.

Thank you for your comment. We have already searched the literature, but we will consider adding this database to our upcoming reviews.

For a correct use of the domes, I would like to refer and add this manuscript to references.

Gazia F, Galletti B, Portelli D, Alberti G, Freni F, Bruno R, Galletti F. Real ear measurement (REM) and auditory performances with open, tulip and double closed dome in patients using hearing aids. Eur Arch Otorhinolaryngol. 2020 Feb 1.

VERSION 2 – REVIEW

REVIEWER	Derek Hoare University of Nottingham, UK
REVIEW RETURNED	29-May-2020

GENERAL COMMENTS	Many thanks for so fully addressing and responding to all my comments. I have no further comments and look forward to your review on this important question.
---

REVIEWER	Francesco Gazia Unit of Otolaryngology Policlinico G Martino, Messina, Italy
REVIEW RETURNED	15-May-2020

GENERAL COMMENTS	Now the article, in my opinion is ready for publication
---